# Cladocera Responses to the Climate-Forced Abrupt Environmental Changes Related to the Late Glacial/Holocene Transition

Marta Rudna [1,2,*], Marta Wojewódka-Przybył [3], Jacek Forysiak [2], Krystyna Milecka [4] and Daniel Okupny [5]

1   Doctoral School of Exact and Natural Sciences, University of Lodz, Banacha 12/16, 90-237 Łódź, Poland
2   Department of Geology and Geomorphology, Faculty of Geographical Sciences, University of Lodz, Prezydenta Gabriela Narutowicza 88, 90-139 Łódź, Poland
3   Institute of Geological Sciences Polish Academy of Sciences, Twarda 51/55, 00-818 Warsaw, Poland
4   Faculty of Geographical and Geological Sciences, Institute of Geoecology and Geoinformation, Adam Mickiewicz University, Bogumiła Krygowskiego 10, 61-680 Poznań, Poland
5   Institute of Marine and Environmental Sciences, University of Szczecin, Mickiewicza 18, 70-383 Szczecin, Poland
*   Correspondence: marta.rudna@edu.uni.lodz.pl

**Abstract:** This article aims to trace in detail the periods of rapid changes during the Late Glacial period based on a subfossil Cladocera analysis and a palynological, geochemical, and statistical analysis. At the end of the Older Dryas, the water level in the reservoir was low, with quite cold waters and inconvenient conditions for developing Cladocera-dominated cold-tolerant species. The beginning of the Alleröd is marked by increasing vegetation density and a rising water temperature, with favorable conditions for developing rare species. At its end, there was a large diversity of species, along with the quite deep and rather mesotrophic nature of the water body. The beginning of the Younger Dryas is a shift back to conditions similar to those noticed during the Older Dryas. The shift to Holocene is manifested by a rapid increase in the number of species and abundance of planktonic forms that appeared before the Holocene onset. The high resolution of the research (1 cm sampling) allowed us to set up more precisely the boundaries between the stadials and interstadials of the Late Glacial and to find some species which were found in the sediment earlier than in previous studies.

**Keywords:** subfossil Cladocera; high resolution; geochemistry; Late Glacial; Older Dryas/Alleröd transition; Alleröd/Younger Dryas transition; Younger Dryas/Holocene transition; Żabieniec mire





## 1. Introduction

One of Earth science's most discussed research topics is contemporary climate change and related environmental transformations. Global warming, for instance, affects aquatic plants and animals, forcing complex cause–effects chains [1] and altering their habitat. The response of organisms to climate-induced changes is highly variable and, by this, less predictable than physical or chemical environmental parameters. Although substantial progress has been made in understanding aquatic biota responses to climate-related environmental changes, high-resolution and long-term records (e.g., lake sediment cores) are still needed. For instance, the outcomes of a paleolimnological analysis allow us to recognize and reproduce changes in climatic and environmental conditions over past geological periods.

The youngest period in geological history (Quaternary) has been punctuated by the glacial–interglacial cycles. According to Milankovic's theory, these cycles last about 100,000 years, and about 90,000 fall on glaciation [2]. Glacial intervals, however, are heterogeneous, and paleoclimatic records may distinguish phases of warmer and colder conditions.

In Poland, the Last Glacial Maximum (LGM) occurred about 21–20 ka BP years ago [3] and has been archived in many paleorecords. In the Łódź region (Central Poland), for instance, the LGM was identified in sediment profiles spanning the period from the Middle Weichselian to the Kamion Phase—corresponding to the Epe Phase in Denmark [4,5]. The transition from the Late Glacial (the youngest part of the last glaciation) to the Early Holocene ranged from about 17,000 to about 8200 years ago and was marked by abrupt climate fluctuations, manifested by alternating cold (stadials) and warm (interstadials) periods [5–9]. These rapid climatic alterations were related to the transitions between Oldest Dryas/Bölling, Bölling/Older Dryas, Older Dryas/Alleröd, Alleröd/Younger Dryas, and Younger Dryas/Early Holocene, and they varied in their amplitude. Consequently, the course of climate changes over the shift from the Late Glacial to the Holocene was very complicated, as evidenced in ice-core or palynological records [10,11]. This is particularly well seen in Greenland's ice cores, for which oxygen isotope analyses were performed. The constructed oxygen curves are considered to reflect global (thermal) climate changes [12]. It is noteworthy that the last glacial–interglacial climate variability also had numerous environmental consequences for inland waters, encompassing their abiotic and biotic factors. Cold–dry stages, for instance, induced the lake level to drop, leading to the rearrangement of biota structure (often simplifying it), with the domination of cold-tolerant species. In contrast, warmer stages promoted more species-rich biota assemblages [13–17].

The time intervals of a highly dynamic climate, such as those reported during the Late Glacial/Holocene transition, are especially interesting because they may constitute a model of current and potential future climate changes and address the questions about the origins of environmental changes during past climate events. Therefore, analyzing past climate oscillations may help us (1) to understand and explain the causes of the current changes occurring in nature, (2) to distinguish between natural appearing climate changes and human-induced ones, (3) to answer the question of whether past climate modifications are analogous to current ones, and (4) to understand in depth and determine ecosystems resilience and resistance to climate-induced environmental disturbances. Such knowledge is crucial for predicting future changes in freshwater ecosystems forced by climate warming and mitigating their negative consequences. A high-resolution sampling of sediment cores can provide detailed information on the course of environmental changes during climate events [18]. Such an approach seems particularly important for paleorecords spanning the Late Glacial/Holocene transition [16]. Unfortunately, previous analyses often involved point sampling of sediment cores at a resolution of 4 to 10 cm, which does not display the continuity of changes, meaning that the reconstructed picture is incomplete. In such an approach, the difference in age between successive samples can be even 100 or 200 years. Such resolution is insufficient in the case of abrupt climate changes, such as during the Late Glacial/Holocene transition. For instance, transitions from cold to warm conditions last much shorter; for example, the climate shift during the transition between Oldest Dryas and Bölling lasted only a few years [10]. The newest research of the sediments of the Lake Gościąż based on microfacies, isotope analysis, and paleoclimatic reconstructions showed that the temperature reduction at the beginning of the Younger Dryas lasted 180 years, while the final warming lasted only 70 years [19]. Consequently, rapid climate changes may push freshwater ecosystems into a new mode. It acts on the biota structure, productivity, and function of water bodies. Therefore, palaeoecological studies of the last transition from a glacial to an interglacial have analyzed various fossil organisms.

One of the most abundant and diverse aquatic groups—and, thus, one of the most useful to infer the past climate and environmental conditions—is Cladocera, also called water fleas [20]. These microcrustaceans inhabit almost all types of freshwater ecosystems in pelagic and littoral zones and react quickly to environmental changes. Because their chitinous exoskeletons are usually well preserved in sediments, they are valuable proxies for reconstruction purposes [21]. These tiny animals are susceptible to such environmental factors as water depth [22], pH [23], trophy state [24], and temperature [25]. Consequently, water fleas' subfossil remains can track long-term environmental changes. The analysis of

subfossil Cladocera remains works very well in paleoclimatic and paleoecological studies, as proved by numerous research conducted worldwide. For instance, in Central Poland, such studies were undertaken for the sites Koźmin Las [26], Żabieniec [27], Rąbień [28], and Ner–Zawada [29] and revealed a significant alteration within Cladocera fauna that was both directly and indirectly related to climate variability.

This article presents new high-resolution data from the Żabieniec mire (Central Poland) obtained within the revisited study. An analysis was performed at the 1 cm resolution for the newly collected core. Re-examining scientifically important study sites at a higher resolution and/or using new methods is becoming a more common approach for providing new data and/or more accurate results (e.g., Lake Gościąż [19]).

Herein, we focused on the three sections of the 12 m long profile (from depth 940 to 970 cm, 895 to 925 cm, and 765 to 799 cm), assuming that they correspond to periods of rapid climatic and environmental changes. The central part of the research constituted analyses of subfossil Cladocera remains at the 10–20-year resolution (sampling every 1 cm). A subfossil Cladocera analysis was supplemented with a palynological and geochemical analysis and statistical techniques. An appropriate numerical analysis allowed us to track the relationship between environmental factors and Cladocera data during the Alleröd/Younger Dryas and Younger Dryas/Holocene transitions. This approach is novel, contrary to the previous study of the Żabieniec site.

The study's objectives were (1) to examine the role of local habitat conditions in shaping species composition and its succession during rapid climate changes and (2) to provide precise and high-resolution data on how lake and peatland basins responded to the rapid climate changes during the Late Glacial/Holocene transition. The last reason for undertaking the research was (3) to verify whether the sampling density would provide more accurate information on the nature of changes within the reservoir and its surroundings compared to studies in the lower resolution.

## 2. Materials and Methods

### 2.1. Study Site

Żabieniec mire is located in Central Poland (Figure 1), within a glaciogenic landscape, and was created in the melt-out basin, about the kettle-hole shape [30,31]. The depression of the mire is located in the central part of the melt-out area, where over a dozen small buried depressions were documented [30]. Glacial and glaciofluvial hills surround this area, with slopes reaching nearby the bog. There is a watershed between the Mrożyca river and the Mroga river basins. The lithology of bog-buried depression is well-known [30,32]. The lower part of the filling is lake deposits: sand with slit and gyttja layers. The upper section is peat, and lacustrine sedimentation started in the early Late Weichselian. Previous research has shown that Cladocera remains were present in the sediment from the end of the Plenivistulian to Holocene [15].

In mire depression, biogenic sediments of all the Late Glacial and Holocene were deposited, with continuous accumulation. Lacustrine deposits accumulated up to the middle of the Subboreal period [9], and the following were deposited peat layers [31].

Żabieniec mire is in the middle of a closed denudational system that was active during the Late Glacial, including the Younger Dryas. Intensified mechanical denudation processes were predominantly active, combined with wind operation [33,34]. Long primarily sandy and silty slopes of glacial forms surrounding the depression were subject to denudation. The material resultant in this process formed covers in lower parts of slopes and became part of the lacustrine filling-silt, and occasionally very fine sand, as a mineral admixture in gyttija, filled the Żabieniec kettle-hole.

### 2.2. Coring and Subsampling

The deposits of Żabieniec mire in the coring spot Z-2 (Figure 1) consist of sequences of peat (0–380 cm) and gyttja (380–1240 cm), so the general thickness of biogenic deposits is about 12.4 m [33]. The core (Z-3) from the central part of the mire was collected by using

a Więckowski piston corer in 2018, and, afterward, it was divided and subsampled in a laboratory into 1 cm slices for proxy analyses.

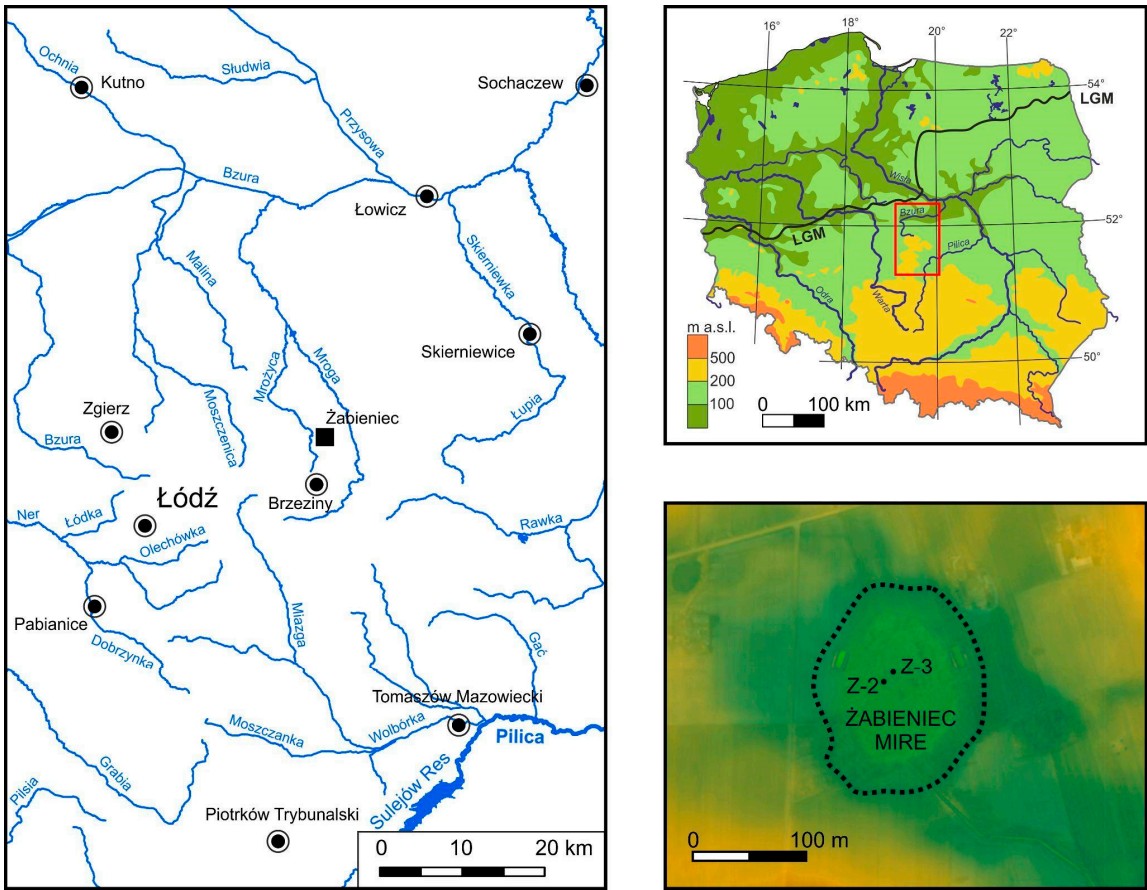

**Figure 1.** Locality of the study site (Żabieniec mire).

### 2.3. Geochronology

The Żabieniec sediment core chronology was established based on 14C AMS in the Gliwice Radiocarbon Laboratory (GdA) and the Analytic Testing Laboratory (Beta). Age determinations were carried out on twelve samples, including eleven macroremains of terrestrial plants and one sample of pollen extract. Conventional radiocarbon dates were calibrated by using OxCal 4.2.2 [35] and the IntCal20 calibration curve [36]. A detailed methodological approach was presented in the paper of Petera-Zganiacz et al. [32].

### 2.4. Cladocera Analysis

A subfossil Cladocera analysis was performed for the three sediment sections (765–799 cm, 895–925 cm, and 940–970 cm), with a resolution of 1 cm (overall 96 samples). Samples were prepared according to the standard methodology of Frey [20]. In total, 1 cm$^3$ of fresh sediments was deflocculated in hot 10% KOH. The residue was washed through a 38 μm sieve, and the volume was topped up to 10 mL with distilled water. The final solution was dyed with safranine, and one-to-five slides (0.1 cm$^3$ each) were analyzed by using a biological microscope for each sample at 100×, 200×, and 400× magnification. All Cladocera remains were counted: headshields, shells, postabdomens, postabdominal claws, and ephippia. A minimum of 100 individuals were counted in each sample [37]. Ephippia were not included in the total Cladocera sum. Species identification was made by following Szeroczyńska and Sarmaja-Korjonen [38]. The results are presented on the total frequency and relative species abundance diagrams, including the ratio of planktonic and littoral cladoceran taxa, and were plotted by using C2 graphics software [39].

### 2.5. Geochemical Analysis

The geochemical analysis procedure for the core section corresponding to the Alleröd/Younger Dryas and Younger Dryas/Holocene transitions was described in detail in the most recent paper of Petera-Zganiacz et al. [32]. For the sediment sequence spanning the Older Dryas/Alleröd transition, only the bulk sediment elemental composition was measured at the 2 cm resolution to determine total carbon (TC), total nitrogen (TN), and total sulfur (TS). TC, TN, and TS were determined by using a VarioMax CNS elemental analyzer. The TOC content was calculated by subtracting TIC (as a results 0.27*LOI925) from TC [40]. Furthermore, the organic matter and calcium carbonate ($CaCO_3$) content were determined by thermogravimetric loss-on-ignition (LOI) analyses, at 550 °C and 925 °C, respectively [41].

### 2.6. Palynological Analysis

For the pollen analysis, 1 $cm^3$ of sediment was taken at a 2–4 cm resolution. Samples were prepared according to the standard procedure [42], as described in detail in the paper of Petera-Zganiacz et al. [32].

### 2.7. Statistical Analysis

A Hierarchical Cluster Analysis (HCA), using Paleontological Statistics (PAST) software version 4.03, was performed to define the border between interstadial and stadial phases in the Cladocera record. The Bray–Curtis index was used to measure the similarity of the abundance distribution of communities. The remaining numerical analysis (DCA, PCA, and RDA) was performed by using R software (packages: factoextra, FactoMinerR, and vegan; functions: PCA, fviz_pca_var, decorana, rda, ordiR2step) [43]. Geochemical data were standardized by using the z-score method. The Shapiro–Wilk test was run to test the normality of geochemical and lithological indices (grain size) and displayed the non-normal distribution of some parameters. The Detrended Correspondence Analysis run on the cladoceran data (z-scored Cladocera abundance; rare species with relative abundance < 5% and single occurring were excluded) displayed a short gradient length (<2) [44]. Therefore, the Principal Component Analysis (PCA; correlation matrix) was used to identify the main changes in the cladoceran assemblages. Data were further synthesized by using the redundancy analysis (RDA) on the standardized dataset. The analysis was run for the sediment section spanning the Alleröd/Younger Dryas and Younger Dryas/Holocene transitions. Grain size, organic matter, mineral matter, and terrigenous silica contents, together with geochemical indicators (Na, K, Ca, Mg, Fe, Mn, Cu, and Zn), were used as the explanatory variables in RDA. These sediment characteristics were broadly described in a paper by Petera-Zganiacz et al. [32].

## 3. Results

### 3.1. Chronology

Radiocarbon dating displayed the Z-3 core span the entire Holocene and a section of the Late Glacial. The lowermost sediment layer dated (943–944 cm) was deposited ca. 13,886 ± 82 modeled cal years BP (Supplementary Table S1). The age-depth model developed for the Z-3 core by Petera-Zganiacz et al. [32] was adapted herein (Figure 2). The palynology biostratigraphy is in agreement with the results of the [14]C dating. Stratigraphy of the older sediments (below 945 cm) was determined based on pollen data and correlation with the Z-2 core [15,27,31,45]. Based on the geochronology, three sections of the sediment sequence were selected for further proxy analysis.

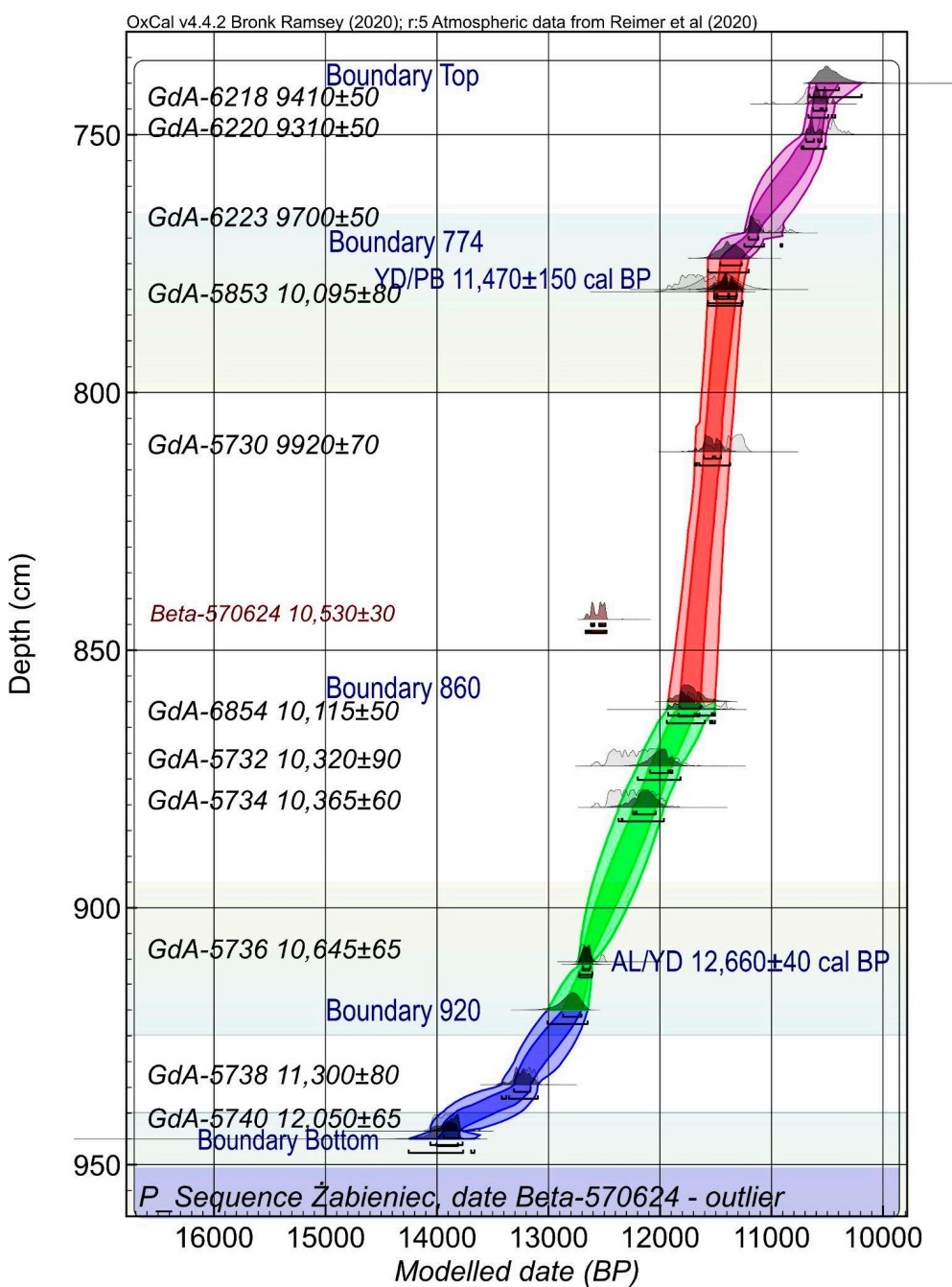

**Figure 2.** Age–depth model established for sediment sequence of Żabieniec mire (followed Petera-Zganiacz et al. [32]). Light blue boxes mark the studied sediment sections [36].

*3.2. Subfossil Cladocera Assemblages*

A total of twenty-five Cladocera species were recorded in the Żabieniec sediment profile. These belong to four families: *Daphnidae*, *Bosminidae*, *Chydoridae*, and *Sididae*.

- Older Dryas/Alleröd transition (970–940 cm)

The cladoceran-inferred biostratigraphic boundary between the Older Dryas and Alleröd was established at the point (957 cm depth) of the significant shift in the Cladocera assemblage and HCA (Figure 3). Nine Cladocera species were identified at the end of the Older Dryas. Subsequently, diversity increased up to twelve taxa, together with the onset of the Alleröd. Over this transition period, the total Cladocera abundance varied from 3433 ind/cm$^3$ (963 cm depth) to 19,500 ind/cm$^3$ (953 cm depth), with slightly higher

values reported for the Alleröd interstadial. Planktonic cladocerans were poorly represented (*Daphnia longipina* group), and littoral taxa dominated the Cladocera assemblage. In particular, *Chydorus sphaericus* and *Alonella nana* occurred throughout this section of the sediment profile and had, on average, 47.8% and 30.5% contributions to cladoceran assemblage, respectively. A consistent decrease of *Ch. sphaericus* and a gradually increasing share of *A. nana* were observed. *Acroperus harpae*, *Alona quadrangularis*, and *Alona guttata* were also largely encountered. Moreover, on average, a higher number of ephippia was recorded compared to the following sediment layers.

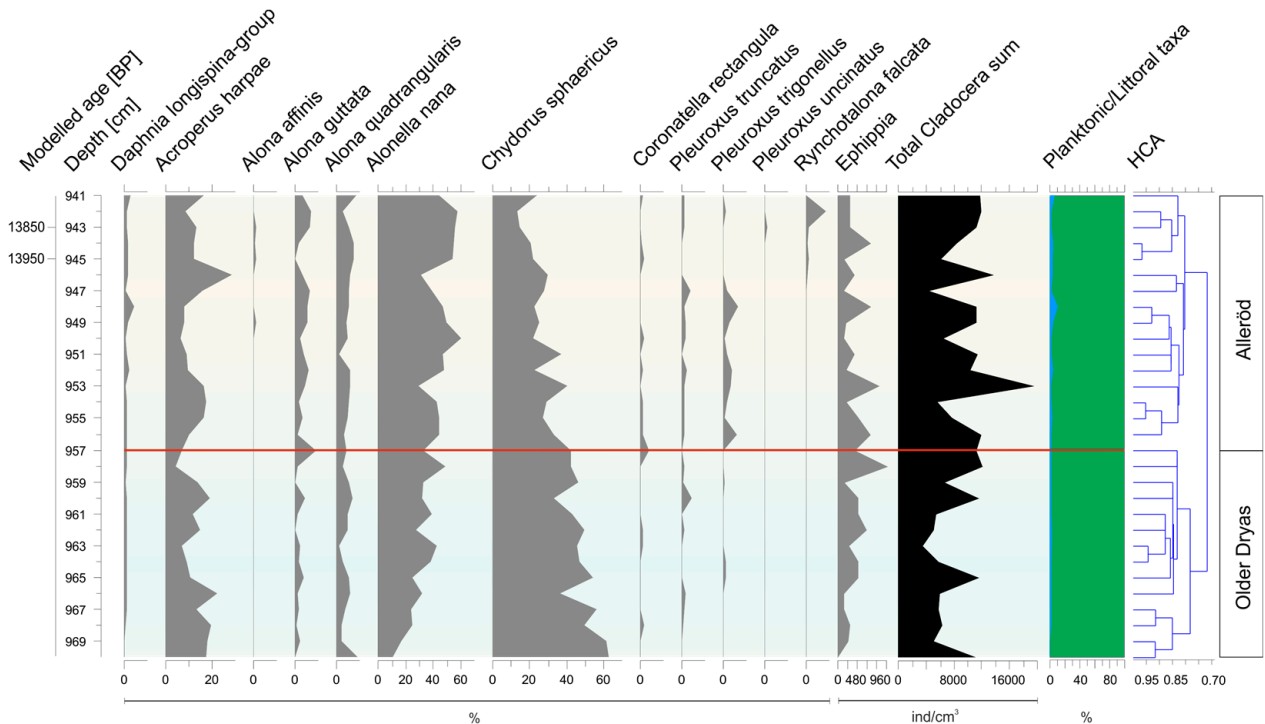

**Figure 3.** Diagram of percentage composition and the total sum of Cladocera specimens in sediments of the core Z-3 determined for Older Dryas/Alleröd transition. HCA: Hierarchical Cluster Analysis.

- Alleröd/Younger Dryas transition (895–925 cm)

The cladoceran-inferred biostratigraphic boundary between the Alleröd and Younger Dryas was defined at a depth of 911 cm (Figure 4). The total number of cladoceran species identified was eighteen. The highest Cladocera concentration was recorded at a depth of 921 cm (20400 ind/cm$^3$), whereas the lowest abundance was noted at a depth of 899 cm (3433 ind/cm$^3$). A clear decreasing trend was observed in the cladoceran abundance from interstadial to stadial. Initially, the assemblage of limnetic species increased significantly and constituted 41% of cladoceran fauna (end of the Alleröd). *Eubosmina* taxa contributed the most to the assemblage of planktonic species. Subsequently, the share of planktonic taxa decreased (up to 14%) and generated a drop in the total Cladocera concentration. Amongst the littoral cladocerans, *A. nana* dominated with a relative abundance of up to 70%, whereas the contribution of *Ch. sphaericus* was significantly lower in comparison to that during the Older Dryas/Alleröd transition (up to 16%).

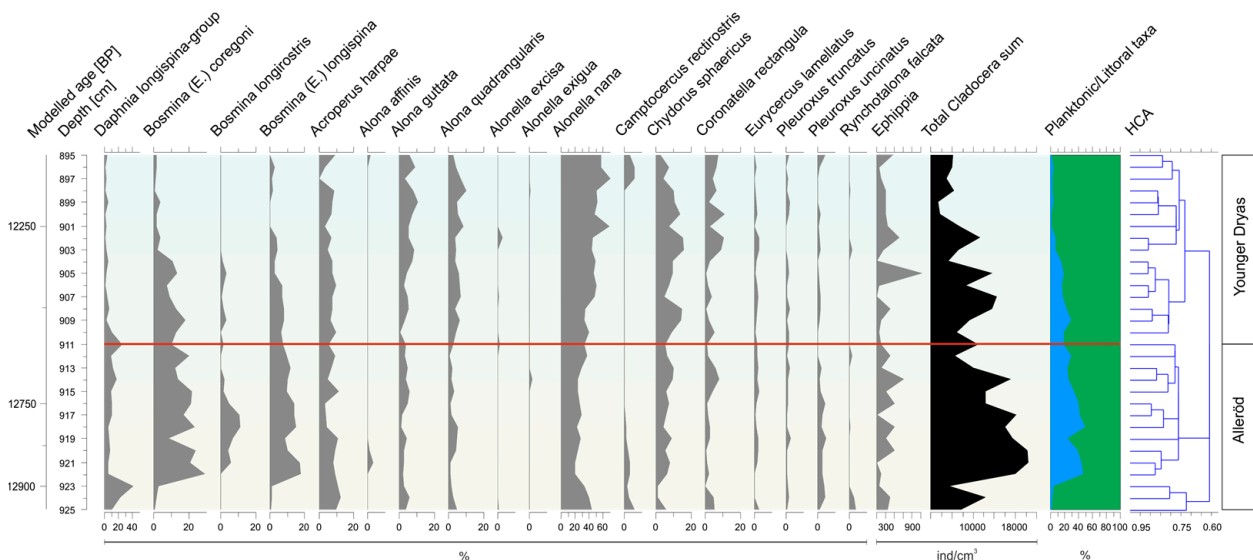

**Figure 4.** Diagram of percentage composition and the total sum of Cladocera specimens in sediments of the core Z-3 determined for Alleröd/Younger Dryas transition. HCA: Hierarchical Cluster Analysis.

- Younger Dryas/Holocene transition (799–765 cm)

The cladoceran-inferred biostratigraphic boundary between the Younger Dryas and the Holocene was set at a depth of 782 cm based on visible changes in the Cladocera assemblages and HCA (Figure 5). Total abundance of Cladocera varied from 5250 ind/cm³ (771 cm depth) up to 23,600 ind/cm³ (796 cm depth). The section of the core associated with the end of the Younger Dryas was dominated by littoral species with a share of up to 98%, whereas the total number of species was equal to 19. *A. nana* and *Ch. sphaericus* predominated, with 58% and 13%, on average, respectively. At the onset of the Holocene, the share of planktonic species increased to 32%, on average, and some species reappeared (*Alonella excisa*, *Camptocercus rectirostris*, and *Sida crystallina*). The average contribution of *A. nana* dropped to 21% in favor of *Bosmina (E.) coregoni* (20%). Within this period, the share of *Ch. sphaericus* remained at 13%. At the same time, the share of the remaining single species did not exceed 10%, and ephippia were rarely encountered.

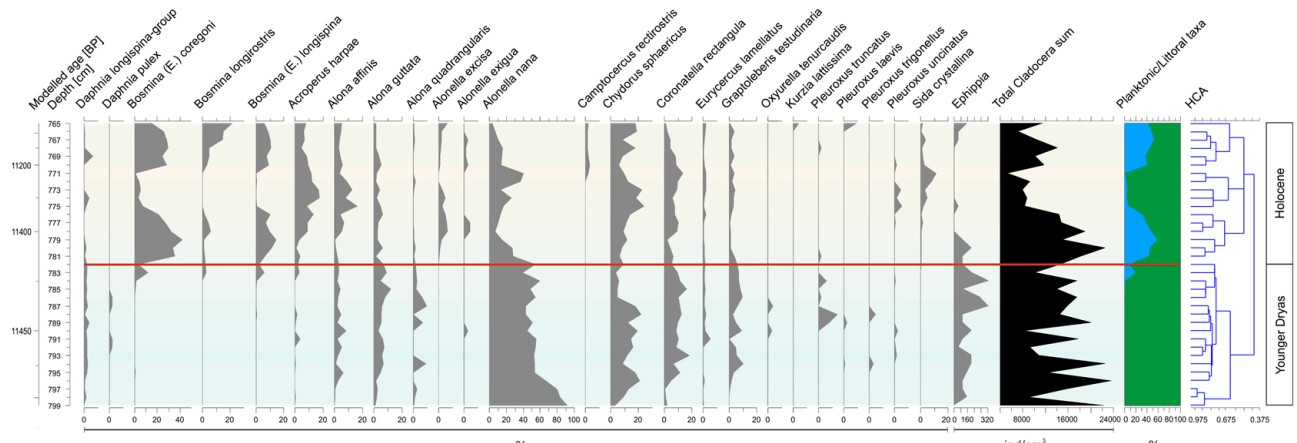

**Figure 5.** Diagram of percentage composition and the total sum of Cladocera specimens in sediments of the core Z-3 determined for Younger Dryas/Holocene transition. HCA: Hierarchical Cluster Analysis.

### 3.3. Geochemistry

The results of the geochemical analysis obtained for the Alleröd/Younger Dryas and Younger Dryas/Holocene transitions have already been described in the studies of Petera-Zganiacz et al. [32] and, therefore, are not discussed in detail herein. It is noteworthy that this recently published work [32] described three lithogeochemical facies (mineral, organic, and mineral–organic) for the sediment section of the Z-3 core spanning the entire Younger Dryas and shortly the pre- and post-periods. The sediments are carbonate free and dominated by silt fraction. The admixture of sand (up to 33%) significantly increased in the deposit of the Younger Dryas/Holocene transition. Moreover, the relatively high Ca, Mg, and Fe content (up to 6, 4, and 13 mg/g) characterizes the Alleröd/Younger Dryas transition. The early Holocene deposits are outstanding due to their high Zn content (up to 170 μg/g) and organic matter.

The new geochemical data were acquired for the sediment sequence section corresponding to the Older Dryas/Alleröd transition. The TOC and TN, together with the S content, display an increasing trend toward the upper part of this section (Figure 6). The TOC/N atomic ratio exhibited a slight fluctuation between 14.45 and 16.94, with lower values at the middle section of the Older Dryas/Alleröd transition, and did not differ significantly from the values reported for sediments corresponding to the Alleröd/Younger Dryas and Younger Dryas/Holocene.

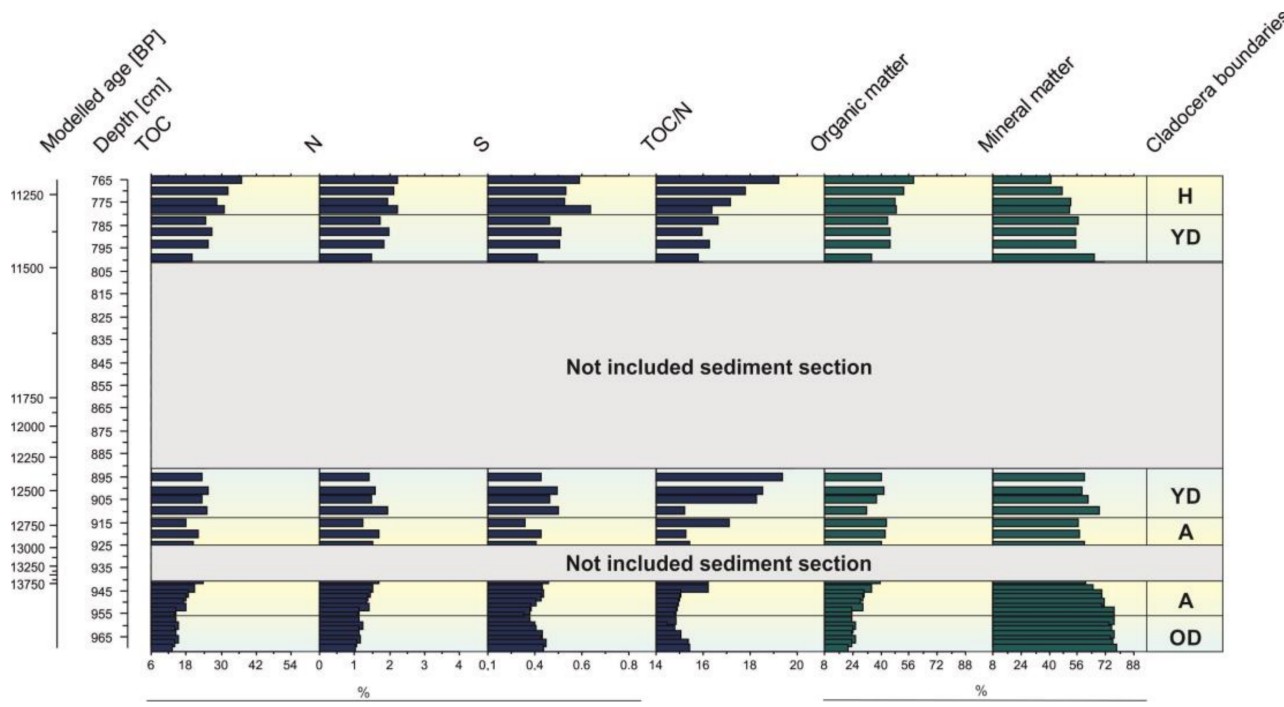

**Figure 6.** Basic geochemical parameters determined for the core Z-3. Abbreviations: OD = Older Dryas, A = Alleröd; YD = Younger Dryas; H = Holocene.

### 3.4. Pollen Record

The border Older Dryas/Alleröd is not very clear. The proportion of AP/NAP does not change very much, and the AP (trees and shrubs) value increases only slightly. The clear change is visible in proportions of *Betula* (decrease, 31% to 20%) and *Pinus* (increase, 54% to 67%), meaning increasing forestation. At the same time, the *Poaceae* and *Ranunculus* type curves decrease as evidence of the shrinking open areas (Figure 7). The curve of the "cold" species *Betula nana* type is changeable but decreases at the beginning of the Alleröd. *Juniperus*, a species typical for cold periods of the Late Glacial, is decreasing also. There is high content of *Pediastrum* in the end of the Older Dryas (exceeding 200% of calculation sum), and it decrease at the beginning of the Alleröd.

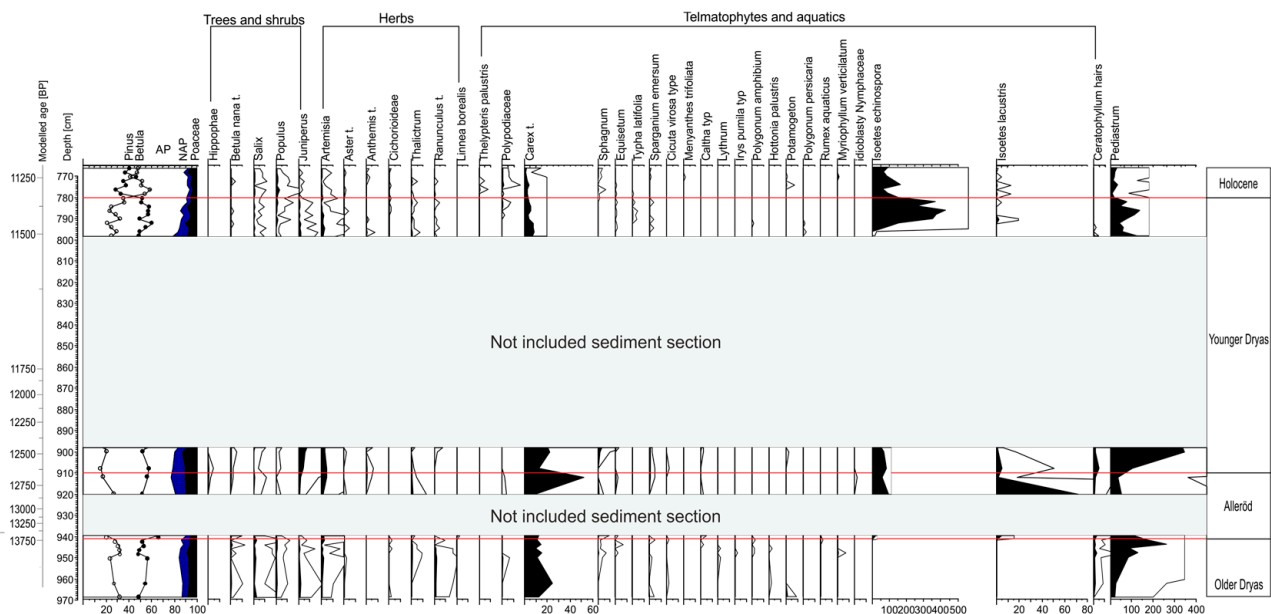

**Figure 7.** Percentage pollen diagram determined for selected sediment sections of Z-3 core.

At the border Alleröd/Younger Dryas, the *Pinus* curve decreases (58% to 53%), and the *Betula* one increases (15% to 21%), meaning shrinking of the forest area. The *Juniperus* curve increases clearly (up to 7%). Species typical for cold periods appear: *Betula nana*, growing contemporary at subarctic regions; and *Hippophaë rhamnoides*, a light-demanding plant of open areas. *Carex* type pollen and *Isoëtes* spores decrease clearly (Figure 7). The *Pediastrum* curve increases up to over 300%.

The border Alleröd/Holocene is determined by very profound changes in pollen-type content. *Pinus* decreases (52% to 34%) and *Betula* increases (36% to 55%) exactly at the turn of Al/H (Figure 7). There is also a decrease of *Juniperus* (to 0.4%) and NAP (*Poaceae*, *Artemisia* in it). At the end of the Alleröd, the pollen of *Typha latyfolia* and a lot of spores of *Isoëtes echinospora* were found. The *Pediastrum* curve decreases at the beginning of the Holocene.

### 3.5. Numerical Analysis

The first two PCA axes of the cladoceran assemblages accounted for 56.6% of the variation within the dataset, with Axes 1 and 2 accounting for 35.9% and 21% of the variation, respectively (Figure 8). The PCA biplot displays arrows representing the scores of the fifteen taxa that mostly contribute to newly created components (i.e., Axes 1 and 2). The highest quality of representation by Axis 1 has *Bosmina* spp. and numbers of ephippia, whereas *G. testudinaria*, *A. harpae*, and *A.affinis* contributed the most to PC2 (Supplementary Figure S1). *A. excisa* and *S. crystallina*, heat-demanding species, are strongly positively associated with PC1. The negative PC1-sample scores reflect cold-tolerant taxa (*Ch. Sphaericus* and *A. quadrangularis*) and ephippia, indicators of deteriorated environmental conditions. Therefore, PC1 likely represents changes related to the water-temperature gradient due to climate changes. Low PC2-sample scores correspond to *A. guttata*, *G. testudinaria*, and *C. rectangula*, representing catchment processes and vegetation development that, in turn, could affect the water pH (alkalinity).

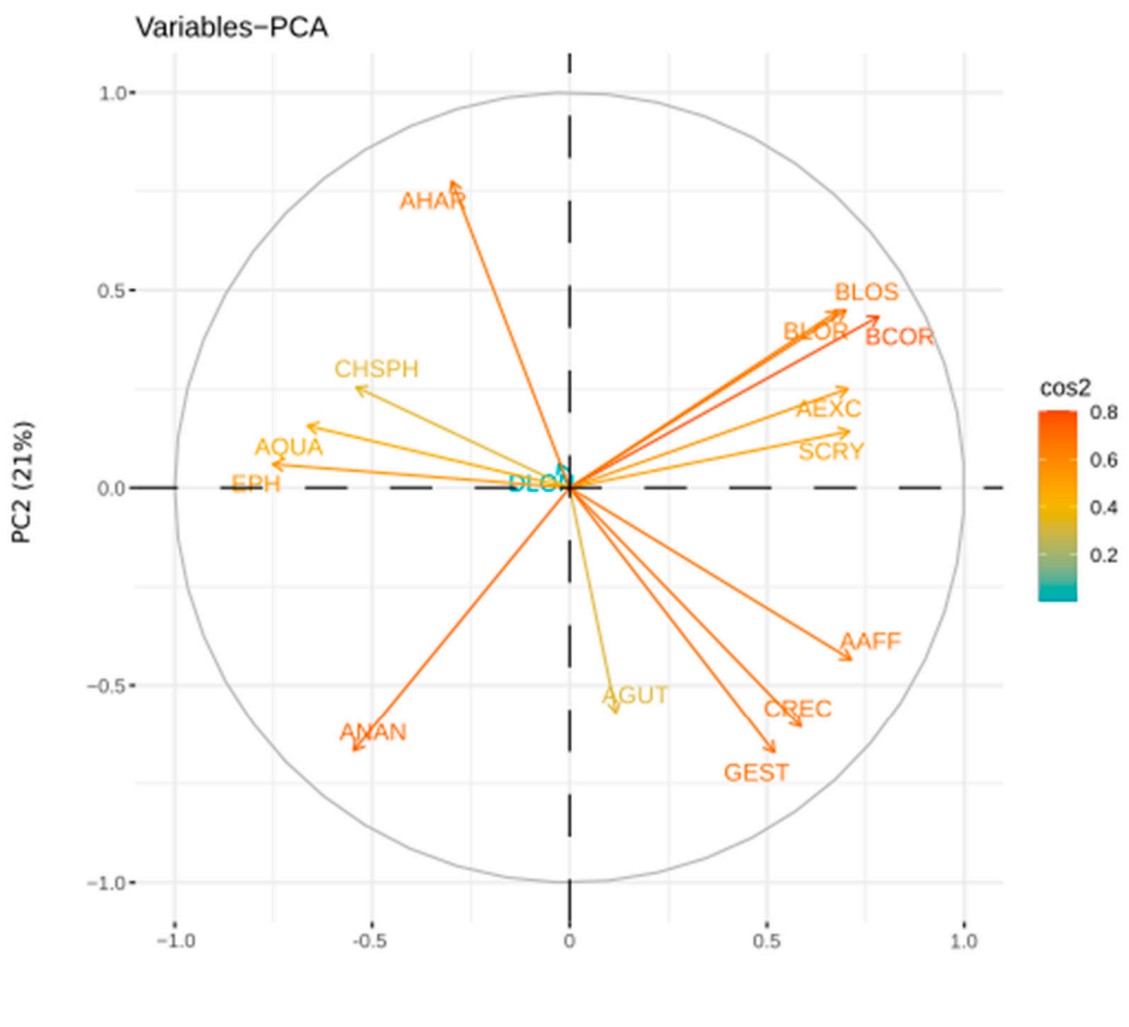

**Figure 8.** Principal Component Analysis (PCA; correlation matrix) Axes 1 and 2 of key cladoceran species. Variable (taxon) names: DLON = *Daphnia longispina* group; BCOR = *Bosmina (E.) coregoni*; BLOR = *Bosmina longirostris*; BLOS = *Bosmina (E.) longispina*; AAFF = *Alona affinis*; AHAR = *Acroperus harpae*; AGUT = *Alona guttata*; AQUA = *Alona quadrangularis*; AEXC = *Alonella excisa*; ANAN = *Alonella nana*; CREC = *Coronatella rectangula*; CHSPH = *Chydorus sphaericus*; GEST = *Graptoleberis testudinaria*; SCRL = *Sida crystallina*; and EPH = ephippia.

The RDA with the forward selection of environmental variables indicates that only three factors (organic matter, Zn, and sand) are statistically significant (Figure 9). The combined effect of the first two canonical axes explains 46.8% of the variance in the cladoceran assemblage at the transition to the Younger Dryas and further to the Holocene. Interestingly, the RDA model shows a higher similarity between the beginning of the Younger Dryas and the end of the Alleröd, so both of these periods link lower organic matter and share of sand fraction. All three arrows may represent a combination of climate change and catchment processes affecting water chemistry. The first axis, correlated with Zn, represents a gradient of water pH with more alkaline (right) to more acid water (left), so the Zn mobility at the sediment–water interface depends on the water pH and redox conditions [46]. Sand and organic matter (OM) arrows reflect intensity of morphogenetic processes (positive values = intensification) and accumulation of organic matter [32]. The latter parameter may be a function of vegetation development and/or trophy state related to the development of peatlands near the Żabieniec site. The positive values of OM (Figure 9) indicate a higher

trophy state. The high trophic level of the lake waters is also represented by the content of biophilic elements, such as S (about 0.5%) and N (often above 1%).

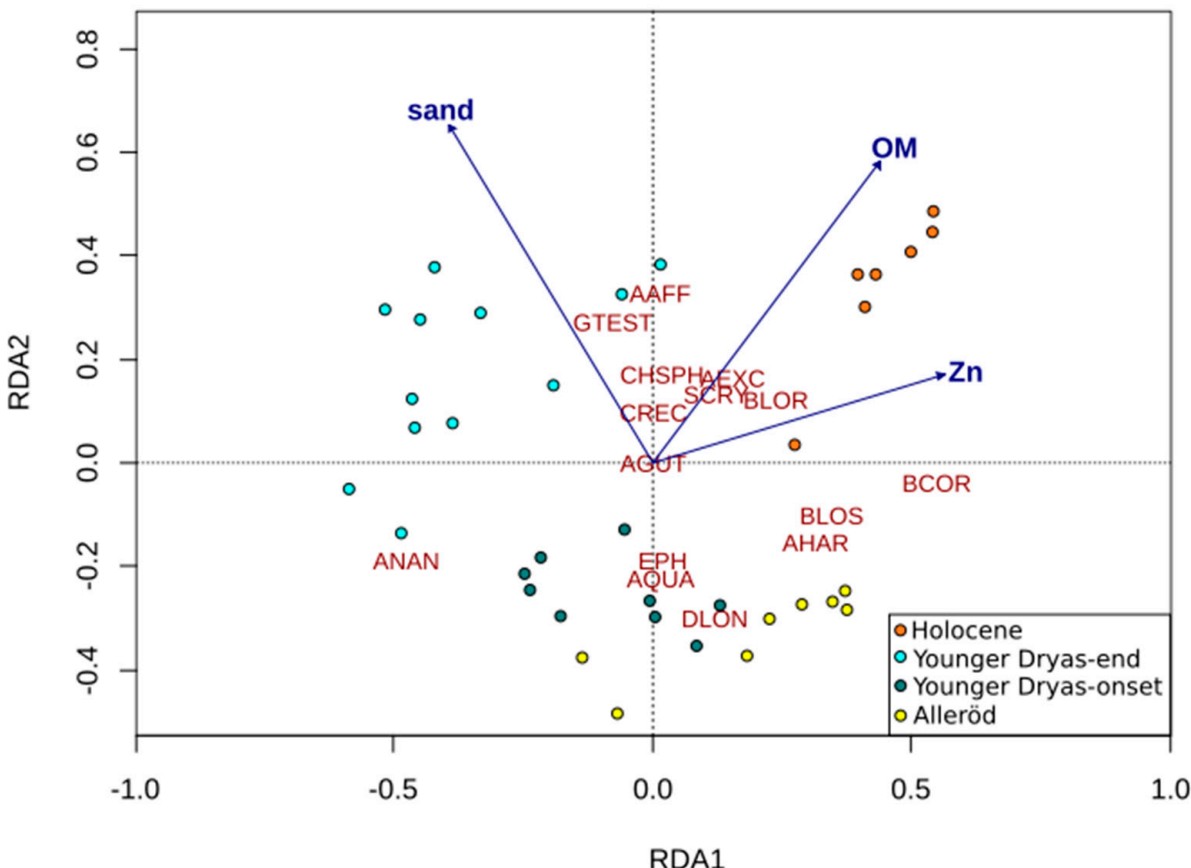

**Figure 9.** Redundancy analysis (RDA) biplot on forward-selected environmental variables, showing the distribution of sediment samples and key cladoceran species. RDA was performed for the Alleröd/Younger Dryas and Younger/Dryas/Holocene transitions. Black arrows indicate the scores of the environmental variable. Colored points indicate locations of sediment samples. Variable (taxon) names: DLON = *Daphnia longispina* group; BCOR = *Bosmina* (*E.*) *coregoni*; BLOR = *Bosmina longirostris*; BLOS = *Bosmina* (*E.*) *longispina*; AAFF = *Alona affinis*; AHAR = *Acroperus harpae*; AGUT = *Alona guttata*; AQUA = *Alona quadrangularis*; AEXC = *Alonella excisa*; ANAN = *Alonella nana*; CREC = *Coronatella rectangula*; CHSPH = *Chydorus sphaericus*; GEST = *Graptoleberis testudinaria*; SCRL = *Sida crystallina*; and EPH = ephippia.

## 4. Discussion

The transition between the Late Pleistocene and Holocene was the period of the profound global transformation of aquatic ecosystems due to climate alteration, especially regarding temperature. The consequences of these were changes in the physicochemical properties of water bodies, thus affecting biota structure. High-resolution paleoecological data, thus, provide valuable information about the long-term ecological impact of climate fluctuation.

The herein reported cladoceran dynamics during the Late Glacial/Holocene transition were manifested mainly in the changes in the share of littoral and planktonic taxa, the changes in ephippia concentrations, and the shift from assemblage dominated by cold-tolerant taxa to more diverse species assemblage with warmth-preferring taxa. The results show that the cladoceran assemblage changes were mainly mediated by climate variation, especially temperature and catchment (vegetation) processes affecting the water pH. This interpretation is displayed in the PCA and RDA results. The positive correlation of *Bosmina*

spp., *S. crystallina*, and *A. excisa* with PC1 indicates their relationship with a higher water temperature. The position of *Bosmina* spp. and *A. affinis* on the PCA biplot, however, reflects the influence of the water pH on their appearance at a similar level as the water temperature. Both factors also affected the abundance of *A. nana*, with its higher share being during colder periods with a lower water pH.

### 4.1. Environment Alteration during the Rapid Climate Changes Related to LGM/Holocene Transition

4.1.1. Older Dryas/Alleröd Transition

The Older Dryas is not always clearly distinguished in stratigraphic schemes [47], as it was a short-lasting period of cooling; therefore, it may be challenging to catch it in the paleorecords. The herein used high-resolution approach, however, should enable us to capture even short-term changes wherein the Older Dryas was characterized by the development of birch forests in the reservoir's surroundings, followed by a reduction in open areas, as indicated by a decrease in the share of *Artemisia* and *Ranunculus* types. The Older Dryas/Alleröd boundary in the pollen record was established ca. 13,706 modeled cal BP (941 cm), which gives several centuries of shift relative to cladoceran signals (before 13,947 modeled cal BP; 957 cm). Thus, the main reason for the discrepancy between cladoceran and pollen records may be the gentle nature of this transition and the small magnitude of short-term changes. It is noteworthy that the cladoceran signals were not clearly marked then, and it was possible to define the Older Dryas/Alleröd boundary by using statistical techniques. Likewise, there is no pollen evidence of significant climate change. The fluctuating share of pine and birch may reflect unstable environmental conditions at that time.

In the Cladocera record, the end of Older Dryas was characterized by the dominance of littoral species from the *Chydoridae* family, which, together with the absence of planktonic taxa, indicates a low water level in the Żabieniec waterbody [48]. The dominant species at that time was *Ch. sphaericus*, which, when it occurs as one of the dominant species, is considered an indicator of high eutrophy [49]. On the other hand, this species is eurytopic, highly adaptive, and often a pioneer in lakes [50]. *Ch. sphaericus* has been found to dominate the cladoceran assemblages during cold intervals with a too-harsh climate for other taxa. Therefore, the predominance of *Ch. sphaericus*, together with other taxa from the group of so-called "arctic species", such as *A. harpae* and *A. affinis* [50–52], indicates rather cold and unfavorable conditions for Cladocera development. This is further supported by relatively low species diversity and a fairly large number of resting eggs (ephippia). They are produced in response to environmental deterioration, e.g., a rapid decrease in water temperature [53], and by this, they are useful as indicators of stress conditions. Consequently, there is a visible decrease in the share of *Ch. sphaericus* in favor of *A. nana* during the transition to the Alleröd. It seems, therefore, that *Ch. sphaericus* was outcompeted by *A. nana*. The latter species is associated with the macrophyte (e.g., *Potamogeton* and *Chara*) in the littoral zone [50]. Regarding the PCA, the significant share of *A. nana* and *A. guttata*, which are species tolerating a pH as low as ca. four and preferring oligosaprobic waters [54], may indicate a slightly acidic character of the water [49]. The appearance of the thermophilic *Pleuroxus trigonellus* and *Pleuroxus uncinatus* also points to increasing vegetation density and rising water temperature [54,55]. There is a lack of clear evidence for climate warming in the pollen record, but the slightly lower flux of *Salix* may suggest higher temperatures. Furthermore, the decline in the total number of ephippia (associated with low values of PC1), together with the increase in the total sum of Cladocera, further supports the amelioration of environmental conditions in the Żabieniec waterbody and more favorable conditions for Cladocera assemblage development. At the same time, the water level is still low, as suggested by the lack of planktonic taxa, but the trophy state could slightly increase. The higher nutrient content is signalized by the appearance of *Pediastrum* and the high trophic index of Fe/Ca that was noted in the previously studied core, Z-2 [45]. However, the high values of the latter index were explained by the birch

influx [45], whereas the presence of *Pediastrum* may also be climate-mediated. For instance, Rundgren [56] found a high concentration of *Pediastrum* during mild periods. Additionally, turbidity due to the intensification of erosion processes may alter water transparency and limit light penetration, thus directly influencing aquatic plants [57]. The gradual decrease of mineral matter content in the Z-3 core toward Alleröd, therefore, allows us to assume the increasing water transparency (lower turbidity), which contributed to the aquatic vegetation flourishing, is realistic, and it may explain the higher *Pediastrum* content. However, lower mechanical erosion at the onset of the Alleröd contradicts the geochemical analysis from the Z-2, which points to lower catchment erosion, as indicated by Na+K+M/Ca [45]. However, these analyses were performed at a 5 cm resolution and represent rather a long-term scale of changes than the short-term alteration we are focused on herein. Furthermore, the theory about a slightly higher trophy state cannot be excluded certainly, but its significant increase is debatable. Higher Cladocera abundance may result from improved temperature and food availability due to higher input of nutrients. The latter hypothesis is supported by slightly higher values of N and TOC in the sediments. The high TOC/N indicates that the organic matter is a mixture of aquatic and terrestrial biomass. Therefore, higher TOC/N at the beginning of the Alleröd, contrary to the end of the Older Dryas, may suggest a more substantial contribution of carbon produced by vascular plants (higher input of terrestrial organic matter after Meyers and Teranes [58]). On the other hand, a significantly higher proportion of TOC and, in turn, higher TOC/N may result from climatic shifts toward slightly higher humidity. This change triggers the enhancement of allochthonous carbon transportation [59]. There is no additional evidence for that theory; therefore, higher organic matter results likely from higher vegetation density inside the lake and on its edge.

The cladoceran reaction to a general cooling in the Older Dryas is widely described in the literature [51,60–62], and our results do not differ significantly from the previous findings. The species composition of cladocerans in the sediments of lakes in Northern and Central Poland was dominated mainly by a few cold-tolerant taxa (e.g., *Ch. sphaericus*, *A. harpae*, *A. nana*, and *A. affinis*) [15,27], and the number of individuals was relatively low. Similar observations come from mountain lakes in Romania [14] and the Czech Republic [63]. In some lakes of the Łęczna-Włodawa Lake District (Southeast Poland), there were noted planktonic taxa (*Bosminidae*, *Daphnidae*), suggesting deeper initial water levels, contrary to Żabieniec and other northern located sites [16]. Furthermore, species diversity is higher toward the southeast position of waterbodies in Poland. The relatively diverse Cladocera assemblage in the Żabieniec site was explained by Pawłowski [27] as being the result of the warmer microclimate in this part of Poland. Another factor that should be considered is the distance from the ice-sheet limit [64].

### 4.1.2. Alleröd/Younger Dryas Transition

The end of the Alleröd, contrary to its onset, was marked by large species diversity (18 species), higher total Cladocera sum, and a low number of ephippia. All of this proves that there were favorable conditions for the development of Cladocera. The fauna of the Cladocera is dominated by plankton species, especially from the *Bosminidae* family, indicating relatively deep-water conditions and presumably higher habitat availability, allowing for the flourishing of Cladocera assemblage. The coexistence of species characteristic of oligo-mesotrophic and eutrophic waters and the high share of *Bosmina coregoni* suggest a rather mesotrophic nature of the water body at that time. Thermophilic species such as *Pleuroxus* and *Camptocerus rectirostris* at the end of Alleröd, along with the cold-tolerant *Eurycercus lamellatus* [65], point to a moderately warm climate. In the transition to the Younger Dryas, the total number of Cladocera decreased, the number of ephippia increased, and cold-tolerant species such as *Chydorus sphaericus* and *Alonella nana* became dominant. All of the above suggests the deterioration of abiotic conditions and a shift back to conditions similar to those noted during the Older Dryas. The temperature drop is also reflected in the pollen record by the higher share of *Juniperus* and *Betula nana*, as well as a lower share of *Thalictrum*, which prefers warmer climatic conditions. As a consequence of climate

change, longer ice covering of the lake likely implied oxygen depletion, as indicated by Fe/S [32], which, in turn, affected cladoceran abundance [66]. At the same time, the share of planktonic species significantly declines to reflect a decrease in the water level, which contradicts the water level fluctuation reconstructed by Forysiak [67]. The pollen record also gives an opposite signal—the deepening of the lake that manifests itself in the higher share of *Pediastrum* and appearance of *Potamogeton* with the drop in the *Carex type* and *Polypodiaceae*. An alternative scenario is lowering the water level and/or shrinking the lake surface area and increasing nutrient input. The rise in the concentration of *Pediastrum* and *Potmogeton*, together with the drop in *Isoetes lacustris*, was driven by the shift toward a higher trophy state [68]. A lower water volume could trigger higher nutrient content; however, a more likely factor could enhance catchment erosion, especially since the previous water-level reconstruction suggests an increase at that time [67]. Lower forest density and the domination of open plant communities caused an increase in the supply of mineral matter and catchment-derived nutrients [69]. The *Hippophae* curve signalizes extending open areas, whereas geochemical indices reflect increasing weathered material [32]. Furthermore, considering that the Żabieniec reservoir is a kettle-hole with a narrow littoral zone, it may be assumed that a drop in the water level and/or shrinking of the lake surface area caused a decrease of habitat availability (limited moist habitats). Such environmental transformation could lead to a lowering of the concentration of *Carex* type and *Polypodiaceae*. On the other side, the geochemical indices (TOC/N, Na, Fe, Zn, K, and Mg) and the presence of *Sphagnum* suggest an expansion of peatlands on the lake shore [32]. The higher share of *A. guttata* and *A. nana* indicates a subtle decrease in the water pH, supporting the peatland development near the lake. Considering all the above, we hypothesize that both transformations within cladoceran and pollen assemblages resulted from lateral changes in vegetation type on the lake edge and in the ratio of littoral/open water open zones. Because of the steep wall of the kettle hole, increasing the water level could extend littoral/benthic habitats and increase the littoral taxa's share, whereas the drop in the absolute number of planktonic *Bosminidae* likely resulted from a temperature decrease, as suggested by Pawłowski [15] and supported by PC1. Our theory, however, needs to be further tested by using other biological proxies. It cannot be excluded that the discrepancy between cladoceran and pollen signals is a consequence of the time lag in their reaction; the Younger Dryas was a period of rapid, short-lasting climate changes and uniform environmental conditions [62].

Cladocera species' composition at the Żabieniec site at the end of the Alleröd does not differ significantly from other Polish records [51,62,70–73]. *G. testudinaria* was reported from other Polish sites, contrary to our cladoceran record. The literature shows that cladoceran response to climate changes at the onset of the Younger Dryas varies and depends on the geographic location and depth of the reservoir [55]. Mainly, a decline in planktonic forms in favor of littoral and cold-tolerant species was observed [26,60,74], and this coincides with our findings. The herein-reported water pH decrease was a broader tendency, as expressed in the cladoceran records in Poland [16] and Romania [14], and it would result from the prolonged ice cover.

### 4.1.3. Younger Dryas/Early Holocene Transition

The absence of planktonic species such as *Bosmina coregoni*, *Bosmina longirostris*, and *Bosmina longispina* at the end of the Younger Dryas indicates the relatively shallow water conditions [51] or harsher climatic conditions. The significant share of a highly specialized species such as *Graptoleberis testudinaria* indicates a well-developed plant cover. The species (*Pleuroxus* group, *C. rectirostris*) preferring warmer water conditions have almost disappeared. The increased number of ephippia, again, and low species diversity testify to the deterioration of climatic and environmental conditions unfavorable for the development of Cladocera. Along with the onset of the Early Holocene, the living conditions of Cladocera improved due to the increase in water temperature, as suggested by the appearance of the stenothermic species *S. crystallina* [54]. The improvement of thermal conditions is also

confirmed by the withdrawal of tundra species, the development of pine forests, and the appearance of *Typha latyfolia* [32]. Planktonic species from the *Bosminidae* group reappeared and became dominant, indicating that the water level rose [26]. A palynological analysis suggests that the lake's depth was 3–4 m [32]. However, this hydrological condition seems unstable, as reflected by the variable share of planktonic taxa. *Typha latyfolia* also signals water-level fluctuation. The water pH presumably increased, and this is indicated by a lower share of acid-tolerant species, such as *A. nana* and *A. guttata* [55]. Moreover, a drop in the share of *G. testudinaria* in favor of *A. harpae* was observed. Both species are often associated with aquatic vegetation [54], and such a shift within Cladocera assemblage may result from altering macrophytes' structure and/or the expansion of habitats associated with the sandy bottom. *A. harpae* also willingly inhabits such substratum [54], even at great depths. At that time, the sandy fraction constituted a relatively high proportion of sediments, whereas the pollen record suggests the development of shallow-water communities, i.e., rushes, *Sparganium emersum* type, *Typha latifolia*. A low curve of the *Sphagnum* appears, reflecting, once again, the development of a peat bog nearby.

An abrupt increase in the number of species and specimens along with the Holocene onset has been broadly reported from Polish lakes, e.g., the lakes Ostrowite, Gościąż, Imiołki, and Przedni Staw [62]. The general trend displayed low cladoceran abundance at the end of the Younger Dryas and a rapid increase in the share of planktonic taxa from *Bosminidae* family, heat-demanding species, and those indicating improvement of environmental conditions at the beginning of the Holocene [16,26,48,55,74,75]. The cladoceran pattern reported at the Żabieniec site fits very well with this general tendency observed in most sites. Similar immediate cladoceran responses to Holocene climate warming were reported from Norway [76], or Denmark [47], and seem to have taken place regardless of the geographical location [55].

Cladocera remains from the Late Glacial have been documented from several study sites (Rąbień and Ner-Zawada) [28,29]. These studies, however, include an analysis at lower resolutions (4–10 cm) and/or represent different environmental conditions. For instance, the site of Koźmin-Las is located in a vast river valley [26]. The aforementioned studies [26,28,29] were performed at the 4–10 cm resolution and documented from 9 to 23 species in the entire cores, whereas the Cladocera analysis with 1 cm resolution for the Żabieniec site displayed higher diversity (up to 25 species).

*4.2. Comparison of Low-Resolution and High-Resolution Cladocera Analysis*

The lower-resolution analysis of sediment cores may miss some signals during the abrupt climate changes of the Late Glacial/Holocene transition because of sediment compaction, a relatively low sedimentation rate, and the often high dynamic of biota assemblages [16]. Previous Cladocera analyses for the Żabieniec site (core Z-2) were performed at the resolution 4–10 cm [15,27,77]. The lower-resolution data show that the Older Dryas was characterized by low total Cladocera abundance (1300/cm$^3$) and was dominated by species that were resistant to changing ecological conditions [27]; this partially agrees with our findings. However, data from the core Z-2, manifested the relatively fast withdrawal of *Ch. sphaericus* from the lake at the end of the Older Dryas. Our data displayed the Older Dryas/Alleröd transition as gradual. Both cladoceran and pollen records from the Z-3 core show a small amplitude of environmental changes and a lack of strong signals of transformation. Together with the Alleröd warming, thermophilic species appeared, which is consistent in both cores. The higher-resolution analysis of the Z-3 core allowed us to detect the earlier presence of *P. trigonellus*, since the Alleröd onset. Moreover, new data detected the presence of the rare species *R. falcata*, which was probably in the lake for a long time in low numbers or appearing sporadically under favorable conditions. That species was not documented in samples analyzed at a 5 cm resolution from the Z-2 core [15], whereas a 4 cm resolution analysis performed by the same author [77] shows the occasionally appearance of *R. falcata*. The previous data and newly provided information indicate the essential role of the high-resolution analysis in biogeographical studies.

The Alleröd ending is manifested by the rapid drop in the share of planktonic taxa and the disappearance of *E. lamellatus* in the lower-resolution Cladocera record. Data from the Z-3 core indicate the gradual character of the aforementioned changes in cladoceran assemblage. The subsequent Younger Dryas cooling is archived in the Z-2 core as a period of abrupt environmental changes reflected by the significant drop in the number of Cladocera species (up to five) and total Cladocera abundance. Species-richness in the Z-2, however, was relatively high at the very beginning of the Younger Dryas. When comparing cladoceran records, according to this, they seem not to differ significantly. A slight discrepancy is observed in the number of species that is higher in the Z-3 core (ca. 18), and some differences in the species compositions are documented. *R. falcata* previously reported from the Z-3 core appeared again at the end of the Alleröd and the onset of Younger Dryas. It is noteworthy that Pawłowski [15] identified two species, *Kurzia lattissima* and *Leydigia leydigi*, at the Alleröd/Younger Dryas transition in contrast to our data. It seems that these taxa were reported only in one sample in meager amounts. This discrepancy may suggest that these taxa were occasionally present. Thus, it seems that even with the high-resolution analysis, we may not be able to catch all taxa if we study (1) remains only from one core site and (2) there is a limited volume of the sediment sample. On the other hand, the herein analyzed sediment sequence may not overlap the Z-2 sequence for which *K. lattissima* and *L. leydigi* were reported. This explanation cannot be excluded, as we cannot correlate both cores perfectly.

The shift to Holocene seems to be similarly recorded in both cores and manifested by a rapid increase in the number of species and planktonic forms [15]. However, there are a few differences visible. The Z-3 data indicate the appearance of *Bosmina* species already at the very end of Younger Dryas, whereas in the Z-2 core, they were noted for younger sediment layers. A similar time lag is likely related to *S. crystallina*, *A. excisa*, *A. harpae*, and *A. exigua*. However, this inconsistency may result from diverging between cores in depth established for the Younger Dryas/Holocene boundary. The lower-resolution analysis did not establish boundaries precisely. Subsequently, the higher-resolution analysis allowed us to note the earlier appearance of a rare species, *O. tenuicaudis*, and reported the presence of another rare taxon, *P. laevis*. The latter species has not been found in the Z-2 core at all.

## 5. Conclusions

The sediment record of Żabieniec mire indicates that the cladoceran assemblage responded sensitively mainly to temperature and, secondarily, to water chemistry (water pH and/or alkalinity) mediated by catchment processes. The high-resolution cladoceran data provide information about the nature of climate-induced environmental transformation during the transition from the Late Glacial to the Holocene. The Older Dryas/Alleröd transition had the gentlest character, reflected in the unclear signals of changes in cladoceran and pollen records. The Older Dryas cooling is reflected by the dominance of so-called "arctic species". A substantial increase of *A. nana* at the onset of Alleröd probably resulted from the water pH. The predominance of *A. nana* suggests persisting moderately cold conditions compared to the end of Alleröd. However, climatic and environmental conditions had been ameliorated slightly, as displayed by cladoceran and pollen records. The Alleröd end was characterized by a moderately warm climate and mesotrophic water with a higher pH. There was discrepancy between cladoceran and pollen records regarding water-level changes. The Younger Dryas beginning brought deterioration of climate conditions (temperature drop), affecting Cladocera assemblages and the water pH drop, allowing *A. nana* to dominate again. The onset of the Younger Dryas and the end of the Alleröd in terms of cladoceran assemblage and environmental conditions seem similar, as shown by the RDA model. The change in the structure of Cladocera was especially pronounced during the transition from the Younger Dryas to the Holocene, indicating the most profound and abrupt environmental alteration. The Holocene beginning was distinguished by a higher water temperature and likely trophy, with an increasing water pH and extension of plant cover. Both Cladocera and pollen signalized unstable hydrological conditions.

The higher resolution of the paleolimnological analysis provides us with more accurate information on reservoir development and local environmental changes, leading to a better understanding of the climate effect. Thanks to the high resolution of the sampling, it was possible to capture differences in species composition and Cladocera frequency during specific periods of climate oscillation and identify species that were not found at all or reported at other phases (later) in lower-resolution research. Thus, high-resolution analyses are essential for biogeographic studies of short periods.

**Supplementary Materials:** The following supporting information can be downloaded at https://www.mdpi.com/article/10.3390/w15020348/s1. Figure S1: Diagrams of the quality representation for species on the PC1 (A) and PC2 (B). A high value of cos2 indicates a good representation. Table S1: Results of radiocarbon dating together with calibrated and modeled ages.

**Author Contributions:** Conceptualization, M.R., M.W.-P. and J.F.; methodology, M.R., M.W.-P., K.M. and D.O.; investigation, M.R., M.W.-P., J.F., K.M. and D.O.; writing—original draft, M.R. and M.W.-P.; writing—review and editing J.F., K.M. and D.O.; visualization, M.R., M.W.-P., K.M. and D.O.; supervision, J.F. All authors have read and agreed to the published version of the manuscript.

**Funding:** This research was funded by the National Science Centre in Poland from the funds conferred for research project UMO-2016/21/B/ST10/02451 and statutory resources of the University of Lodz.

**Institutional Review Board Statement:** Not applicable.

**Informed Consent Statement:** Not applicable.

**Data Availability Statement:** Not applicable.

**Acknowledgments:** The authors would like to express their thanks to hab. Joanna Petera-Zganiacz for providing material for research and hab. inż. Danuta J. Michczyńska for sharing the age-depth model.

**Conflicts of Interest:** The authors declare no conflict of interest.

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
