# Peer review of "Cladocera Responses to the Climate-Forced Abrupt Environmental Changes Related to the Late Glacial/Holocene Transition"

_water, doi:10.3390/w15020348_

Round 1
Reviewer 1 Report
The study Cladocera responses to the climate-forced abrupt environmental changes related to the LGM/Holocene transition presents new data on the previously studied site Å»abieniec in Central Poland. The main focus – Cladocera responses to abrupt environmental change, was studied in detail based on the newly obtained Z-3 core. The twelve 14C dates (with only 1 outlier) provide adequate time control.
General remarks:
The article should be more stand-alone/autonomous. In its current form it requires constant looking up to the Peter-Zganiacz et al. 2022 for its full understanding. As the input for statistical analysis are used data, that are not in the paper under review, so it is impossible to verify the correctness of the analysis and further inference fully.
Moreover, the authors assume that readers are familiar with previous works of Pawłowski, regarding the Cladocera of Z-2 core, so skips the basic information about the timespan of Z-2 and if Cladocera remains were present before Older Dryas (in Z-3 below the 969 cm).
Lack of explanation why only AL/YD and YD/PB transition were analysed in detail results in the impression that the research was poorly designed. To avoid this accusation the reasoning for that strategy should be clarified.
Detail remarks (some additional remarks I’ve put directly in the .pdf version of the manuscript):
1. Introduction
It is worth mentioning that the study revisits the previously studied site, so the approach may be similar to GoÅ›ciąż case (Ralska-Jasiewiczowa et al. 1998, Müller et al. 2021) or … (other examples where an important site is studied with higher resolution and/or new methods).
Highlight why the study is novel?
2. Material and Methods
Figure 1. The locality of the study site (Å»abieniec mire) – the figure is very general and does not specify the location Z-2 and Z3 cores.
2.2. Coring => Coring and subsampling
Please specify the respective positions of the Z-2 and Z-3 cores in relation to each other (e.g. a figure similar to Fig. 2 in the paper Petera-Zganiacz et al. 2022). As (in paragraph 4.2) you compare in detail the results of previous studies it should be clear what was the mutual location of both cores Z-2 and Z-3, as well as if there were any methodological differences (e.g. in sample preparation, identification of the remains and presentation of the results). Only when this will be clarified the results may be confronted with each other to draw conclusions that will be attributed to the high/low-resolution approach.
142-145: The deposits of Å»abieniec mire in the coring spot consist of sequences of peat (0 – 380 cm) and gyttja (380 – 1240 cm), so the general thickness of biogenic deposits is about 12,4 m [34]. The core Z-3) from the central part of the mire was collected using a Wieckowski piston corer in 2018 afterward, was divided and subsampled in a laboratory into 1 cm slices for proxy analyses.
2.3. Geochronology
Where possible please include more information instead of a simple redirection to Petera-Zganiacz et al. 2022. Unfortunately, Petera-Zganiacz et al. 2022 is not an open-access paper, so not all of the readers can look into the very important pillar of these studies. Reliable age determination results are crucial to increase the credibility of this study. Is palynology biostratigraphy in agreement with 14C results? If so you may add the statement (referring to Petera-Zganiacz et al. 2022 paper).
2.4. Cladocera methods
What was the minimum counting sum for Cladocera remains?
Ephippia – clarify, that these were/were not included in the general total Cladocera sum for percentage calculation, but were calculated relative to the total Cladocera sum.
164-165: The results are presented on the total frequency abundance and relative species abundance diagrams plotted using C2 graphics software [39].
Planktonic/Littoral ratio is a commonly used form of data presentation for Cladocera, nevertheless it should be mentioned in the Methods section.
2.7. Statistical analysis
Specify which R packages and functions were used.
Please clarify why OD/Allerod transition was not included in RDA
189: “analysis (…) run on cladoceran data” - There should be a more detailed description of the input data (percentages? no taxa were excluded and even taxa of single occurrences were analysed? Ephippia were used as input data, equally to taxa?).
3. Results
3.1. Chronology
Make it clear that the chronology is based entirely on radiocarbon dates and that there are no discrepancies with pollen biostratigraphy.
Figure 2. On the model mark the section of the core that is within the scope of this study ( e.g. similar to Fig. 5. in Pietruczuk et al. 2022. https://doi.org/10.1016/j.catena.2021.105813)
As there are no 14C data below 945 cm, please explain how the Older Dryas was identified.
3.2. Geochemistry
A short synthesis of previously published geochemical is recommended - e.g. a discerning reader will guess that the sediment is probably carbonate-free, but it is worth pointing out this ( and other ) important conclusion derived from the geochemical results.
Moreover, the geochemical environmental variables used for statistical analysis should be at least listed in the relevant method section 2.5 or 2.7).
3.2.
212: “The border between the Older Dryas and Alleröd was established at the point (957 cm depth) of the significant shift in the Cladocera assemblage and HCA” – this implies that Cladocera results rather than radiocarbon dating or pollen was the basis of the stratigraphy. Moreover – the palynological border on fig. 7 is at 940 cm (957 cm at Fig. 3). Please indicate that it is only the “Cladoceran-inferred biostratigraphic boundary”.
4. Discussion
I recommend broader comparative reference to other Cladocera studies within the Late Glacial/Holocene timespan, especially from sites situated outside the range of LGM. Comparison of species richness, composition, and total abundance. This may be useful to highlight that the resolution of the studies for Żabieniec is exceptionally high.
Language remarks:
In its current form, the style and language make the article difficult to read – these may be improved.
Specific language comments:
core portions => core sections
277: growing forestry => increasing forestation
298: significant (statistical implication of the term) => profound
413-414: “mild periods” – what does it mean?
551: more gentle => consider a terminological change – gradual?

Reviewer 2 Report
Shorter, more effective paragraphs need to be presented so that the idea of ​​each thing is easy to grasp.
It is necessary to ensure that each paragraph has only one idea, so that it is more informative. It needs a more consistent sequential presentation, starting from the method, such as mentioning the layers of coring results being analyzed. It would be better if presented sequentially, starting from the lowermost to the last layer to be observed.
